# OSAQ: Outlier Self-Absorption for Accurate Low-bit LLM Quantization

## Abstract

Large Language Models (LLMs) have demonstrated remarkable capabilities in generation and understanding tasks. However, their massive parameter scale leads to significant resource consumption and latency during inference. Post-training weight-only quantization offers a promising solution by reducing model size and accelerating token generation through alleviating the memory-bound issue. Nevertheless, the presence of inherent systematic outliers in weights continues to be a major obstacle. While existing methods, such as scaling and rotation, attempt to address this issue, overall performance remains unsatisfactory. In this paper, we propose Outlier Self-Absorption Quantization (OSAQ), which performs additive weight suppression guided by the second-order low-rank property for low-bit weight-only quantization of LLMs. Specifically, we observe that the Hessian exhibits low-rank consistency across different inputs, with certain directions consistently showing vanishing curvature. Leveraging this property, we identify a stable null space of the Hessian and then construct an additive weight transformation by linearly combining the vectors within this null space, thereby suppressing weight outliers without affecting the task loss. This additive transformation can be absorbed into the weights offline, requiring no inter-layer transformations and introducing no inference overhead. Moreover, the construction is efficiently achieved by a closed-form solution, without resource-intensive training or iterative procedures. Extensive experiments across models of different scales and tasks demonstrate that OSAQ effectively suppresses outliers and enhances low-bit quantization performance. For instance, in 2-bit quantization, OSAQ, when integrated with GPTQ, achieves over 40% lower perplexity compared to vanilla GPTQ.

## 1 Introduction

Large Language Models (LLMs) exhibit exceptional understanding and generation capabilities (Zhao et al., 2023; Zhang et al., 2022; Touvron et al., 2023), achieving state-of-the-art performance across a wide range of complex tasks such as reasoning, knowledge integration, and multimodal interaction. Despite these advances, their success comes at the expense of enormous computational and memory demands (Xiao et al., 2023; Frantar & Alistarh, 2023), which translate into high deployment costs, significant inference latency, and large energy consumption. These inefficiencies pose serious challenges for real-world applications in resource-constrained or latency-sensitive environments, restricting the accessibility of LLMs (Zhou et al., 2024). As a result, reducing model complexity has become a topic of extensive research and growing interest.

Post-training quantization (PTQ), which discretizes model parameters into low-precision values without re-training or fine-tuning, is a promising direction for model compression (Nagel et al., 2020; Li et al., 2021). In the context of LLMs, the decoding process is often constrained by the memory wall (Gholami et al., 2024; Yuan et al., 2024), which severely limits memory access efficiency. Weight-only quantization has thus been widely investigated as an effective solution to this bottleneck (Wan et al., 2023; Lang et al., 2024). Nevertheless, model weights typically contain systematic outliers with large magnitudes, which significantly hinder quantization performance, particularly in low-bit settings. To this end, a variety of approaches have been proposed. For instance, GPTQ (Frantar et al., 2022) compensates for quantization errors of outliers by leveraging approximate second-order Hessian information. AWQ (Lin et al., 2023) mitigates the effects of outliers by

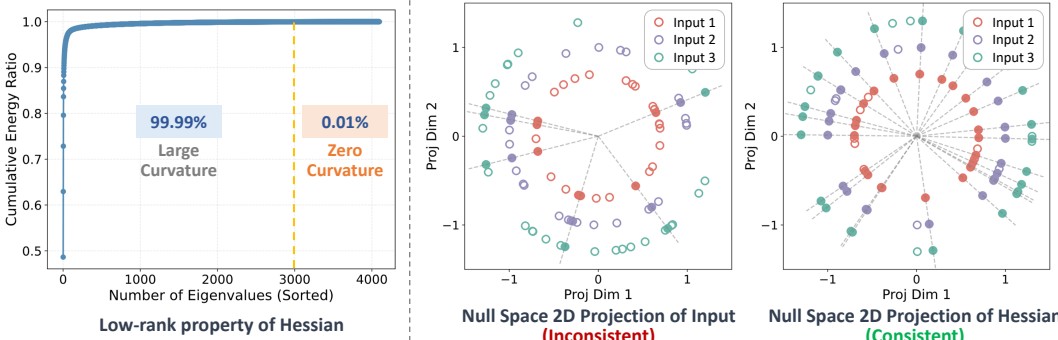

Figure 1: Visualization of the low-rank consistency of Hessian, using data from layer0.attn.k_proj module in LLaMA2-7B. The *left* panel shows that a number of tail eigenvalues together contribute only about 0.01% of the total energy, indicating a pronounced low-rank property. The *right* panel compares the null space of the input and Hessian, where the high-dimensional vectors are projected into a 2D space. It can be seen that although the directions within the input null space vary significantly across samples, the directions within the Hessian null space remain consistent.

scaling the weights according to activation distribution features. QuIP (Chee et al., 2023) introduces an orthogonal rotation of weight matrices to further suppress the presence of outliers. On this basis, several works have further invested in handling outliers within the scaling (Shao et al., 2023) and rotation (Ashkboos et al., 2024; Liu et al., 2024) paradigm.

Despite these efforts, the performance of low-bit quantization is still far from satisfactory, suggesting that relying solely on a multiplicative transformation paradigm is fundamentally inadequate. Therefore, we are motivated to explore *whether there exist additional strategies for suppressing outliers beyond multiplicative transformations between adjacent layers*. Fortunately, we observe that the Hessian of the task loss with respect to a given layer's weights exhibits low-rank consistency across different inputs, as shown in Figure 1; in other words, the Hessian consistently shows negligible curvature in certain directions, referred to as the null space (Strang, 2022). This observation suggests the possibility of applying an additive transformation to the weights while keeping the second-order perturbation of task loss to zero,

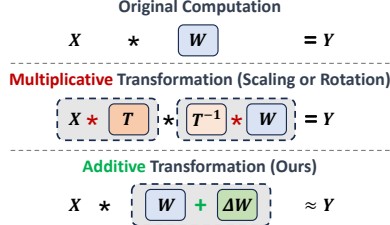

Figure 2: Illustration of additive transformation, which requires no adjustment to other layers while remaining approximately equivalent.

as shown in Figure 2. And more importantly, if such an additive adjustment can counteract the effect of outliers, it provides an effective strategy for suppressing them.

With the above insights, we propose Outlier Self-Absorption Quantization (OSAQ), an accurate low-bit weight-only quantization method for LLMs. OSAQ introduces an *additive weight transformation* beyond previous multiplicative approaches, offering a complementary and scalable scheme. Specifically, for a given layer, we first estimate its approximate Hessian and extract its null space by eigenvalue decomposition. Subsequently, the vectors in the null space are linearly combined with coefficients to construct an additive transformation of the weights. Finally, we minimize the numerical range of the transformed weights by solving for the optimal combination coefficients, thereby effectively suppressing outliers. Note that enabled by the Softmax-$\infty$ objective approximation, the solution of coefficients is obtained *directly in closed form*, without requiring resource-intensive training or iterative procedures. Our main contributions are summarized as follows:

- We propose an outlier self-absorption method that performs additive transformations by exploiting the low-rank properties of the second-order Hessian. It does not rely on inter-layer transformations and is complementary to previous scaling or rotation methods, further improving the performance of low-bit quantization for LLMs.

- The additive transformation is constructed by linearly combining the vectors in the null space of the Hessian. Thanks to the Softmax-$\infty$ objective approximation, the combination coefficients can be derived in closed form to minimize the numerical range of weights.

- Extensive experiments are conducted on various models on a variety of tasks, and OSAQ significantly outperforms the existing methods in low-bit quantization. For instance, in 2-bit quantization, OSAQ achieves over 40% lower perplexity compared to vanilla GPTQ.

## 2 RELATED WORK

**Post-Training Quantization.** Model quantization compresses neural networks by representing floating-point parameters with low-bit values (Gholami et al., 2022; Krishnamoorthi, 2018). While many approaches adopt quantization-aware training (QAT), which requires retraining on the full dataset to achieve strong performance (Choi et al., 2018; Esser et al., 2019), the QAT process is computationally expensive and time-consuming. In contrast, PTQ needs only a small number of samples for calibration, offering a more practical solution for efficient deployment. Previous PTQ methods, such as DFQ (Nagel et al., 2019), AdaRound (Nagel et al., 2020), and BRECQ (Li et al., 2021), perform well on small models, but their effectiveness diminishes on LLMs, where parameters exhibit pronounced outliers that amplify quantization difficulty and accumulate errors as model size grows. This has motivated increasing efforts to develop PTQ schemes for LLMs.

**Weight-Only Quantization for LLMs.** The PTQ methods for LLMs are categorized into two types: weight-activation quantization and weight-only quantization. The former needs to handle outliers in both weights and activations, with notable strategies including scaling (Xiao et al., 2023; Wei et al., 2022), rotation (Ashkboos et al., 2024; Liu et al., 2024), and permutation (Yuan et al., 2023; Lin et al., 2024), which rely on equivalent transformations between adjacent layers. However, the limited precision of activations imposes severe performance bottlenecks. To this end, due to the fact that the decoding efficiency is bounded by the memory access, weight-only quantization has received increasing interest. For outliers in weights, early approaches focused on explicit isolation or iterative compensation. For instance, LLM.int8() (Dettmers et al., 2022) and SqueezeLLM Kim et al. (2023) isolate outliers independently. Afterwards, GPTQ (Frantar et al., 2022) performs error compensation iteratively using approximate Hessian, and MagR (Zhang et al., 2024) iteratively optimizes the infinite norm of the weights. In addition, methods based on equivalent transformations, such as scaling and rotation, have gained wide adoption. Specifically, AWQ (Lin et al., 2023) mitigates the impact of outliers by scaling weights according to activation distributions, and QuIP (Chee et al., 2023) rotates the weights using orthogonal matrices to smooth and suppress outliers.

In this work, we focus on weight-only quantization and aim to explore a novel additive transformation based on Hessian's low-rank consistency as a new strategy for suppressing weight outliers. This approach complements existing inter-layer multiplicative transformations, providing a synergistic scheme that further enhances the performance of low-bit LLM quantization.

## 3 PRELIMINARIES

**Model Quantization.** Quantization discretizes model parameters and represents them using low-precision numerical values (Gholami et al., 2022). To achieve this, the uniform quantizer is the most fundamental and hardware-friendly option, which is defined as follows:

$$\text{Quant: } \boldsymbol{w}^{(\mathbb{Z})} = \text{clip}\left(\left\lfloor\frac{\boldsymbol{w}}{s}\right\rceil + z, 0, 2^b - 1\right), \quad \text{De-Quant: } \hat{\boldsymbol{w}} = s\left(\boldsymbol{w}^{(\mathbb{Z})} - z\right) \approx \boldsymbol{w} \qquad (1)$$

where $\boldsymbol{w}$ and $\boldsymbol{w}^{(\mathbb{Z})}$ denote the floating-point and quantized values, respectively, while the de-quantized value $\hat{\boldsymbol{w}}$ can approximately recover $\boldsymbol{w}$. Here, $\lfloor\cdot\rceil$ represents the rounding operation, and $b$ indicates the quantization bit precision. In this procedure, quantization scale $s \in \mathbb{R}^+$ and zero-point $z \in \mathbb{Z}$ are the key quantization parameters, which are determined by the upper bound $w_{\max}$ and lower bound $w_{\min}$ as follows:

$$s = \frac{w_{\max} - w_{\min}}{2^b - 1}, \quad z = \left\lfloor-\frac{w_{\min}}{s}\right\rceil \qquad (2)$$

It can be observed that the scale $s$ is determined by the numerical range of $\boldsymbol{w}$, which directly affects the quantization resolution. However, in LLMs, the weights exhibit significant outliers, enlarging the range of $\boldsymbol{w}$ and consequently leading to reduced quantization resolution and accuracy.

**Multiplicative Transformation for Outlier Suppression.** To mitigate the outlier issue, various approaches have been proposed. A notable idea is based on scaling (Lin et al., 2023) or rotation (Chee

Figure 3: Workflow of the proposed OSAQ. Under the condition of loss invariance, we explicitly solve for a coefficient matrix to weight the vectors in the Hessian null space, thereby constructing an additive transformation that effectively suppresses outliers in weights.

et al., 2023), which is achieved through equivalent multiplicative transformations between adjacent layers, as follows:

$$(\boldsymbol{X}\boldsymbol{W}_1)\boldsymbol{W}_2 = (\boldsymbol{X}\boldsymbol{W}_1)\boldsymbol{T}^{-1}\boldsymbol{T}\boldsymbol{W}_2 = \boldsymbol{X}(\boldsymbol{W}_1\boldsymbol{T}^{-1})(\boldsymbol{T}\boldsymbol{W}_2) = (\boldsymbol{X}\boldsymbol{W}_1')\boldsymbol{W}_2' \quad (3)$$

where $\boldsymbol{X}$ denotes the input, and $\boldsymbol{W}_1$ and $\boldsymbol{W}_2$ are the weights of two adjacent layers. $\boldsymbol{T}$ represents the transformation, which corresponds to scaling when it is a one-dimensional vector, and to rotation when it is a two-dimensional orthogonal matrix.

Despite certain improvements, the performance of low-bit quantization remains far from satisfactory, indicating that a single multiplicative paradigm is insufficient for handling outliers. Thus, we seek to explore a novel additive paradigm as a complementary approach to outlier suppression.

## 4 OBSERVATIONS AND INSIGHTS

**Observations: Low-Rank Consistency of Hessian.** When the weights undergo a small additive perturbation, the second-order Taylor expansion of the task loss $\mathcal{L}$ for the weights is as follows:

$$\mathbb{E}[\mathcal{L}(\boldsymbol{w} + \Delta\boldsymbol{w}) - \mathcal{L}(\boldsymbol{w})] = \Delta\boldsymbol{w}^T g^{\boldsymbol{w}} + \frac{1}{2}\Delta\boldsymbol{w}^T \boldsymbol{H}^{\boldsymbol{w}} \Delta\boldsymbol{w} + O(\|\Delta\boldsymbol{w}\|^3) \approx \frac{1}{2}\Delta\boldsymbol{w}^T \boldsymbol{H}^{\boldsymbol{w}} \Delta\boldsymbol{w} \quad (4)$$

where $\boldsymbol{w}$ is the flattened vector of weights $\boldsymbol{W}$, $g^{\boldsymbol{w}} = \mathbb{E}[\nabla_{\boldsymbol{w}}\mathcal{L}(\boldsymbol{w})]$ is the first-order gradient, $\boldsymbol{H}^{\boldsymbol{w}} = \mathbb{E}[\nabla_{\boldsymbol{w}}^2\mathcal{L}(\boldsymbol{w})]$ is the second-order Hessian.

As illustrated in Figure 1, we observe that although the low-rank structures of different inputs themselves are inconsistent, the Hessian $\boldsymbol{H}^{\boldsymbol{w}}$ exhibits a pronounced low-rank property. Specifically, along certain feature directions, the corresponding eigenvalues vanish, indicating that their magnitudes are effectively zero. The associated eigenvectors collectively form the null space of $\boldsymbol{H}^{\boldsymbol{w}}$. More importantly, this null space remains stable across different input samples, meaning that the eigenvectors within it do not change. This reveals the low-rank consistency of Hessian.

**Insights: Loss-Invariant Additive Transformation.** By definition of the null space (Strang, 2022), multiplying $\boldsymbol{H}^{\boldsymbol{w}}$ by any vector within the null space yields zero. Therefore, by forming a weighted combination of these null-space vectors, we can construct $\Delta\boldsymbol{W}$. This enables an additive transformation that guarantees the loss remains unchanged:

$$\boldsymbol{W}' = \boldsymbol{W} + \Delta\boldsymbol{W} \quad \text{s.t.} \ \Delta\boldsymbol{w}^T \boldsymbol{H}^{\boldsymbol{w}} \Delta\boldsymbol{w} = 0 \quad (5)$$

where $\Delta\boldsymbol{w}$ is the flattened vector. In this way, although the transformation is not equivalent as in the multiplicative cases, it can still ensure that model performance remains unaffected. Furthermore, with careful optimization, if $\Delta\boldsymbol{W}$ can counteract the influence of outliers, it will serve as a promising strategy for outlier suppression. Since this transformation operates solely on the weights themselves, without adjustment or compensation from the other layers, it is referred to as outlier self-absorption. Figure 3 illustrates the overall workflow of this approach.

## 5 OSAQ: OUTLIER SELF-ABSORPTION QUANTIZATION

Building on the above observations and insights, we aim to construct a low-rank-guided $\Delta\boldsymbol{W}$ to perform an additive transformation on the weights, enabling outlier self-absorption while preserving model performance. Given a weight matrix $\boldsymbol{W} \in \mathbb{R}^{M \times N}$, with $M$ as the output channel dimension and $N$ as the input channel dimension, the construction process of $\Delta\boldsymbol{W}$ is detailed below.

**Extraction of Null Space.** First, following the work (Frantar et al., 2022), to reduce computational complexity, we approximate the Hessian using the input correlation matrix, i.e., $\boldsymbol{H^w} \approx \boldsymbol{X}^T\boldsymbol{X}$. Here, $\boldsymbol{H^w} \in \mathbb{R}^{N \times N}$ is a positive semi-definite matrix. Then, we perform eigen-decomposition on the Hessian $\boldsymbol{H^w}$ and arrange its eigenvalues in non-decreasing order, as follows:

$$\boldsymbol{H^w} = \boldsymbol{V} \operatorname{diag}\left(\lambda_1, \ldots, \lambda_N\right) \boldsymbol{V}^T, \quad 0 \leq \lambda_1 \leq \cdots \leq \lambda_N \tag{6}$$

where $\boldsymbol{V} \in \mathbb{R}^{N \times N}$ is the matrix of eigenvectors, and $\lambda_1, \ldots, \lambda_N$ are the corresponding eigenvalues. Typically, the eigenvalues $\lambda$ are not exactly zero. Using a fixed threshold directly may lead to imbalanced null space dimensions across different layers. Therefore, we adopt a tail-energy strategy, where we start from the smallest eigenvalues and accumulate them to obtain the prefix energy, and the null-space dimension is determined as the smallest $K$ such that the cumulative tail energy reaches a predefined threshold, as follows:

$$\mathcal{N} = \boldsymbol{V}_{[:,0:K-1]}^T, \quad \text{where } K = \min\left\{k : \sum_{i=1}^{k} \lambda_i \geq \gamma \sum_{i=1}^{N} \lambda_i\right\} \tag{7}$$

where $\gamma \in (0, 1)$ is the tail-energy threshold. $\mathcal{N} \in \mathbb{R}^{K \times N}$ denotes the null space of $\boldsymbol{H^w}$, where each row corresponds to a feature direction along which $\boldsymbol{H^w}$ exhibits vanishing curvature.

**Softmax-$\infty$ Objective Approximation.** After obtaining the null space, we introduce a weighting coefficient matrix $\boldsymbol{\beta} \in \mathbb{R}^{M \times K}$ to assign weights to each vector within the null space, thereby constructing $\Delta\boldsymbol{W}$, i.e., $\Delta\boldsymbol{W} = \boldsymbol{\beta}\mathcal{N}$. Our objective is to minimize the numerical range of the weights after applying the additive perturbation. A straightforward approach is to minimize the $\ell_\infty$ norm as follows:

$$\min_{\boldsymbol{\beta}} \|\boldsymbol{W} + \Delta\boldsymbol{W}\|_\infty = \min_{\boldsymbol{\beta}} \|\boldsymbol{W} + \boldsymbol{\beta}\mathcal{N}\|_\infty \tag{8}$$

However, the $\ell_\infty$ norm is non-differentiable, which requires iterative optimization. To address this, we adopt a Softmax-$\infty$ approximation (Boyd & Vandenberghe, 2004), which approximates the original $\ell_\infty$ norm by optimizing the differentiable $\ell_2$ norm of the softmax-scaled values. Specifically, we apply the softmax operation along the output-channel dimension, as follows:

$$s_{ij} = \frac{\exp\left(|W_{ij}|/\tau\right)}{\sum_{t=1}^{N} \exp\left(|W_{it}|/\tau\right)} \tag{9}$$

where $i = 1, \cdots, M$, and $\tau > 0$ is a temperature coefficient. When $\tau$ is large, it captures the average behavior across all components, while as $\tau \to 0^+$, it increasingly emphasizes extreme peak values. In this case, applying the $\ell_2$ norm to the peak-emphasized parameters can serve as an approximation of the $\ell_\infty$ norm, thereby enabling effective identification and suppression of outliers.

**Explicit Solution of Coefficient Matrix $\boldsymbol{\beta}$.** Next, we formulate the optimization objective as a softmax-scaled $\ell_2$ norm. Since the quantization scale and zero-point are both computed along the output-channel dimension, for notational clarity, we explicitly present the $\ell_2$ norm optimization objective for each output channel. The objective for the $i$-th output channel is given as follows:

$$\min_{\boldsymbol{b}_i} \frac{1}{2} \sum_{j=1}^{N} s_{ij} \left(W_{ij} + \boldsymbol{b}_i^T \boldsymbol{n}_j\right)^2 + \frac{\mu_1}{2} \|\boldsymbol{b}_i\|_2^2 + \frac{\mu_2}{2} \left(\boldsymbol{b}_i^T \boldsymbol{v}\right)^2 \tag{10}$$

where $\boldsymbol{b}_i = \boldsymbol{\beta}[i,:] \in \mathbb{R}^K$, $\boldsymbol{n}_j = \mathcal{N}[:,j] \in \mathbb{R}^K$, $\boldsymbol{v} = \mathcal{N}\mathbf{1}_N \in \mathbb{R}^K$, and $\mu_1, \mu_2 > 0$ are the balancing coefficients. In the above optimization objective, the first term is the $\ell_2$ norm, which serves to reduce the numerical range and suppress outliers within each channel; the second term is a regularization term on $\boldsymbol{b}_i$, intended to prevent excessive corrections; the third term imposes a anti-shift constraint, which penalizes uniform translations of the entire channel in the same direction.

By taking the derivative of the objective function with respect to $\boldsymbol{b}_i$ and setting the first-order optimality condition to zero, we obtain the normal equation as follows:

$$\boldsymbol{A}_i \boldsymbol{b}_i = -\boldsymbol{\rho}_i, \quad \boldsymbol{A}_i = \sum_{j=1}^N s_{ij}\, \boldsymbol{n}_j \boldsymbol{n}_j^T + \mu_1 \boldsymbol{I}_K + \mu_2 \boldsymbol{v}\boldsymbol{v}^T, \quad \boldsymbol{\rho}_i = \sum_{j=1}^N s_{ij} W_{ij} \boldsymbol{n}_j \qquad (11)$$

Thus, we can obtain the closed-form optimal solution of $\boldsymbol{b}_i$ under the first-order optimality condition, and by combining all $\boldsymbol{b}_i$, we finally construct the complete coefficient matrix $\boldsymbol{\beta}$ as follows:

$$\boldsymbol{\beta}^* = [\boldsymbol{b}_1^*, \cdots, \boldsymbol{b}_M^*]^T, \quad \text{where } \boldsymbol{b}_i^* = -\boldsymbol{A}_i^{-1}\boldsymbol{\rho}_i, \ i = 1, \cdots, M \qquad (12)$$

**Remark 1.** Each rank-one matrices $\boldsymbol{n}_j \boldsymbol{n}_j^T$ and $\boldsymbol{v}\boldsymbol{v}^T$ is positive semi-definite, and $\mu_1 \boldsymbol{I}_K$ is strictly positive definite. Since $s_{ij} > 0$, $\mu_1 > 0$, and $\mu_2 > 0$, it follows that $\boldsymbol{A}_i$, the second-order derivative of the objective function with respect to $\boldsymbol{b}_i$, is symmetric positive definite and thus invertible, i.e., $\boldsymbol{A}_i \succeq \mu_1 \boldsymbol{I}_K \succ \boldsymbol{0}$. Consequently, $\boldsymbol{b}_i^*$ exists, is unique, and is the unique global minimizer.

# 6 EXPERIMENTS

## 6.1 EXPERIMENTAL SETUPS

**Models and Datasets.** We perform quantization on popular pre-trained LLMs, including LLaMA2 (7B, 13B, 70B) (Touvron et al., 2023), and LLaMA3 (8B, 70B) (Dubey et al., 2024), and larger-scale instruction-tuned LLMs, including Mistral-Large-123B-Instruct (Mistral AI Team, 2024) and Llama-3.1-405B-Instruct (Dubey et al., 2024). We evaluate the performance on language generation tasks using perplexity on WikiText2 (Merity et al., 2016) and C4 (Raffel et al., 2020) datasets, and on commonsense QA tasks using zero-shot accuracy on PIQA (Bisk et al., 2020), ARC (Clark et al., 2018), and WinoGrande (Sakaguchi et al., 2021) datasets. We also assess the quantized models on MMLU (Hendrycks et al., 2020) and MT-Bench (Zheng et al., 2023) benchmarks.

**Baselines.** We compare the proposed OSAQ with several strong weight-only quantization baselines, including GPTQ (Frantar et al., 2022), AWQ (Lin et al., 2023), QuIP (Chee et al., 2023), MagR (Zhang et al., 2024), and OmniQuant (Shao et al., 2023). We also consider quantizing both the weights and KV-Cache, and compare our method with WKVQuant (Yue et al., 2024). In the 2-bit quantization setting, we further enhance performance by incorporating coordinate descent iterations (Behdin et al., 2023), denoted by the † symbol.

**Implementation Details.** We primarily focus on weight-only quantization, while also exploring KV-Cache quantization. The proposed QSAQ serves as a plug-and-play component that complements existing methods to further enhance performance. Accordingly, the quantization setup is aligned with the respective baselines. For example, when combined with GPTQ, the calibration data consists of 128 samples with a sequence length of 2048. For the selection of hyperparameters, we perform a simple grid search over $\gamma \in \{1\text{e-}4, 2\text{e-}4, 5\text{e-}4\}$, $\tau \in \{0.05, 0.1, 0.2\}$, and $\{\mu_1, \mu_2\} \in \{5\text{e-}4, 1\text{e-}3, 2\text{e-}3\}$. It is demonstrated that the quantization performance is robust to the choice of these values, with further details provided in ablation studies.

## 6.2 MAIN RESULTS

**Visualization of weight distributions.** Figure 4 illustrates the weight distributions before and after applying the additive transformation. As shown, the original model contains prominent outliers that enlarge the dynamic range. After applying OSAQ, these outliers are effectively suppressed, resulting in a more compact distribution, which substantially reduces quantization difficulty.

**Evaluation on Language Generation Tasks.** Table 1 reports perplexity results of quantized LLaMA2 and LLaMA3 models. Incorporating OSAQ consistently improves performance over all baselines, with particularly large gains in low-bit settings. In the W3A16 setting, OSAQ+GPTQ reduces perplexity on LLaMA2-13B from 6.44 to 5.72 and on LLaMA3-8B from 13.0 to 11.3, and in the W2A16 setting, OSAQ+GPTQ achieves significant improvements.

**Evaluation on Commonsense QA Tasks.** Table 2 shows that OSAQ consistently improves zero-shot accuracy of LLaMA3 models. In the 4-bit setting, OSAQ brings stable gains over QuIP, main-

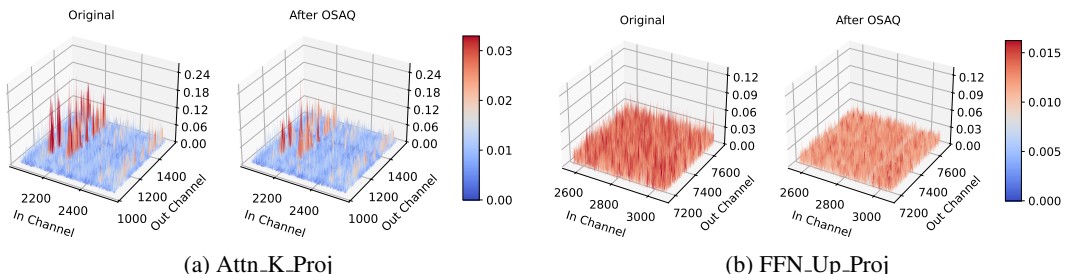

(a) Attn_K_Proj                     (b) FFN_Up_Proj

Figure 4: Weight distributions before and after the additive transformation in LLaMA2-7B's 0-th layer. The original model exhibits prominent outliers, whereas OSAQ effectively suppresses them, substantially reducing the difficulty of quantization. More results are provided in Appendix A.

Table 1: Perplexity results ($\downarrow$) of quantized models on language generation tasks. This table reports the results of LLaMA2 and LLaMA3 models of various scales under different quantization settings.

| Prec. | Method | WikiText2 | | | | | C4 | | | | |
|---|---|---|---|---|---|---|---|---|---|---|---|
| | | 2-7B | 2-13B | 2-70B | 3-8B | 3-70B | 2-7B | 2-13B | 2-70B | 3-8B | 3-70B |
| FP16 | Baseline | 5.47 | 4.88 | 3.31 | 6.10 | 2.90 | 6.97 | 6.46 | 5.52 | 9.20 | 6.90 |
| W4A16 | RTN | 6.11 | 5.20 | 3.67 | 8.70 | - | 7.71 | 6.83 | 5.79 | 14.0 | - |
| | MagR | 5.91 | 5.17 | 3.58 | - | - | 7.52 | 6.81 | 5.72 | - | - |
| | OmniQuant | 5.74 | 5.02 | 3.47 | - | - | 7.35 | 6.65 | 5.65 | - | - |
| | QuIP | 5.94 | 5.01 | 3.53 | 6.50 | 3.40 | 8.01 | 6.88 | 5.87 | 11.1 | 7.10 |
| | AWQ | 6.15 | 5.12 | 3.57 | 7.10 | 3.70 | 7.68 | 6.74 | 5.71 | 10.1 | 7.40 |
| | OSAQ$_{+AWQ}$ | 5.99 | 5.04 | 3.53 | 6.82 | 3.57 | 7.50 | 6.67 | 5.66 | 9.93 | 7.22 |
| | GPTQ | 5.83 | 5.13 | 3.58 | 7.00 | 3.60 | 7.37 | 6.70 | 5.67 | 11.8 | 7.40 |
| | OSAQ$_{+GPTQ}$ | 5.73 | 5.04 | 3.48 | 6.82 | 3.42 | 7.34 | 6.64 | 5.61 | 11.5 | 7.24 |
| W3A16 g128 | RTN | 6.66 | 5.51 | 3.97 | 27.9 | 11.8 | 8.40 | 7.18 | 6.02 | 110 | 22.0 |
| | MagR | 6.46 | 5.45 | 3.95 | - | - | 8.22 | 7.12 | 6.00 | - | - |
| | OmniQuant | 6.03 | 5.28 | 3.78 | - | - | 7.75 | 6.98 | 5.85 | - | - |
| | AWQ | 6.24 | 5.32 | 3.83 | 8.20 | 4.80 | 7.84 | 6.94 | 5.82 | 11.6 | 8.00 |
| | OSAQ$_{+AWQ}$ | 6.08 | 5.23 | 3.75 | 7.96 | 4.61 | 7.75 | 6.89 | 5.72 | 11.4 | 7.87 |
| | GPTQ | 6.29 | 5.42 | 3.85 | 8.20 | 5.20 | 7.89 | 7.00 | 5.85 | 13.7 | 10.5 |
| | OSAQ$_{+GPTQ}$ | 6.07 | 5.30 | 3.79 | 7.98 | 4.99 | 7.77 | 6.95 | 5.78 | 13.4 | 10.2 |
| W3A16 | RTN | 539 | 10.7 | 7.52 | 2.2e3 | 3.2e4 | 402 | 12.5 | 10.0 | 560 | 112 |
| | MagR | 8.66 | 6.55 | 4.64 | - | - | 10.8 | 8.26 | 6.77 | - | - |
| | AWQ | 24.0 | 10.5 | 6.29 | 12.8 | 5.92 | 23.9 | 13.1 | 7.11 | 16.8 | 9.71 |
| | OmniQuant | 6.58 | 5.58 | 3.92 | - | - | 8.65 | 7.44 | 6.06 | - | - |
| | GPTQ | 8.37 | 6.44 | 4.82 | 13.0 | 5.88 | 9.81 | 8.02 | 6.57 | 45.9 | 9.66 |
| | OSAQ$_{+GPTQ}$ | 6.75 | 5.72 | 4.21 | 11.3 | 5.49 | 8.70 | 7.54 | 6.09 | 23.1 | 8.62 |
| | QuIP | 6.50 | 5.34 | 3.85 | 7.50 | 4.70 | 8.74 | 7.34 | 6.14 | 11.3 | 8.00 |
| | OSAQ$_{+QuIP}$ | 6.37 | 5.25 | 3.81 | 7.43 | 4.65 | 8.68 | 7.27 | 6.06 | 10.8 | 7.83 |
| W2A16 g128 | RTN | 1.2e4 | 5.8e3 | 7.8e3 | 1.9e3 | 4.6e5 | 7.2e4 | 3.9e3 | 3.6e3 | 2.5e4 | 4.7e5 |
| | MagR$^\dagger$ | 9.94 | 7.63 | 5.52 | - | - | 14.1 | 10.6 | 8.05 | - | - |
| | OmniQuant | 11.1 | 8.26 | 6.55 | - | - | 15.0 | 11.1 | 8.52 | - | - |
| | GPTQ | 36.8 | 28.1 | 19.2 | 210 | 11.9 | 33.7 | 21.0 | 13.8 | 4.1e4 | 22.8 |
| | OSAQ$_{+GPTQ}$ | 21.2 | 13.4 | 10.7 | 63.8 | 8.36 | 18.3 | 13.8 | 9.44 | 352 | 14.7 |
| | OSAQ$_{+GPTQ}$$^\dagger$ | 10.6 | 7.60 | 5.97 | 26.1 | 6.11 | 14.7 | 10.5 | 8.42 | 62.0 | 10.2 |

taining accuracy close to FP16 baseline. In 3-bit quantization on LLaMA3-8B, OSAQ+GPTQ increases the average accuracy from 63.6% to 65.2%, while on LLaMA3-70B the improvement is from 72.4% to 74.4%, substantially higher than vanilla GPTQ.

**Evaluation on MMLU Benchmark.** Table 3 reports the zero-shot accuracy on the MMLU benchmark for LLaMA2-7B and LLaMA3-8B. The results show that OSAQ consistently improves MMLU accuracy under both 4-bit and 3-bit quantization. For example, OSAQ+GPTQ raises the average score from 38.7% to 39.6% on LLaMA2-7B and from 57.6% to 58.2% on LLaMA3-8B in the 4-bit setting, and also provides 1.0–1.5% gains in the 3-bit case.

**Evaluation of Larger Instruction-Tuned Models.** Table 4 presents quantization results for instruction-tuned models with 123B and 405B parameters. The results show that even at this

Table 2: Zero-shot accuracy (↑) of quantized models on commonsense QA tasks. This table reports the results of LLaMA3 models of various scales under different quantization settings.

| Prec. | Method | LLaMA3-8B | | | | | LLaMA3-70B | | | | |
|---|---|---|---|---|---|---|---|---|---|---|---|
| | | PIQA | ARC-e | ARC-c | Wino | Avg. | PIQA | ARC-e | ARC-c | Wino | Avg. |
| FP16 | Baseline | 79.9 | 80.1 | 50.4 | 72.8 | 70.8 | 82.4 | 86.9 | 60.3 | 80.6 | 77.6 |
| W4A16 | QuIP | 78.2 | 78.2 | 47.4 | 73.2 | 69.2 | 82.5 | 86.0 | 58.7 | 79.7 | 76.7 |
| | OSAQ$_{+QuIP}$ | 78.8 | 78.9 | 48.0 | 73.1 | 69.7 | 82.4 | 86.3 | 60.0 | 80.1 | 77.2 |
| W3A16 g128 | AWQ | 77.7 | 74.0 | 43.2 | 72.1 | 66.8 | 81.4 | 84.7 | 58.0 | 78.6 | 75.7 |
| | OSAQ$_{+AWQ}$ | 78.2 | 75.1 | 43.9 | 72.4 | 67.4 | 82.1 | 85.4 | 58.9 | 79.2 | 76.4 |
| | GPTQ | 74.9 | 70.5 | 37.7 | 71.1 | 63.6 | 80.6 | 79.6 | 52.1 | 77.1 | 72.4 |
| | OSAQ$_{+GPTQ}$ | 76.4 | 72.2 | 39.7 | 72.3 | 65.2 | 81.9 | 82.3 | 54.7 | 78.7 | 74.4 |
| W3A16 | QuIP | 76.8 | 72.9 | 41.0 | 72.5 | 65.8 | 82.3 | 83.3 | 54.9 | 78.4 | 74.7 |
| | OSAQ$_{+QuIP}$ | 77.6 | 73.5 | 41.6 | 72.7 | 66.4 | 82.3 | 84.0 | 55.5 | 78.9 | 75.2 |
| W2A16 g128 | GPTQ | 53.9 | 28.8 | 19.9 | 50.5 | 38.3 | 62.7 | 38.9 | 24.6 | 59.9 | 46.5 |
| | OSAQ$_{+GPTQ}$ | 58.1 | 38.6 | 25.8 | 54.9 | 44.4 | 65.5 | 42.5 | 32.0 | 63.8 | 51.0 |
| | OSAQ$_{+GPTQ\dagger}$ | 64.6 | 44.2 | 37.7 | 60.4 | 51.7 | 69.2 | 49.0 | 39.7 | 68.8 | 56.7 |

Table 3: Zero-shot accuracy (↑) of quantized models on MMLU benchmark. This table reports the results of LLaMA2-7B and LLaMA3-8B models under different quantization settings.

| Prec. | Method | LLaMA2-7B | | | | | LLaMA3-8B | | | | |
|---|---|---|---|---|---|---|---|---|---|---|---|
| | | STEM | Hums | Social | Others | Avg. | STEM | Hums | Social | Others | Avg. |
| FP16 | Baseline | 34.4 | 39.8 | 47.3 | 47.1 | 42.2 | 53.8 | 54.9 | 73.3 | 70.4 | 63.1 |
| W4A16 | OmniQuant | 28.8 | 32.2 | 34.7 | 35.8 | 32.9 | 49.4 | 49.1 | 66.6 | 64.4 | 57.4 |
| | GPTQ | 32.7 | 36.9 | 42.6 | 42.6 | 38.7 | 47.3 | 52.3 | 66.0 | 64.9 | 57.6 |
| | OSAQ$_{+GPTQ}$ | 33.6 | 37.5 | 43.5 | 43.9 | 39.6 | 48.0 | 52.7 | 66.7 | 65.3 | 58.2 |
| W3A16 | OmniQuant | 29.1 | 31.1 | 30.6 | 30.4 | 30.3 | 26.3 | 27.8 | 29.5 | 29.9 | 28.4 |
| | GPTQ | 28.2 | 27.0 | 32.1 | 29.9 | 29.3 | 26.2 | 29.2 | 34.4 | 30.0 | 30.0 |
| | OSAQ$_{+GPTQ}$ | 29.3 | 29.4 | 33.8 | 31.1 | 30.9 | 27.0 | 30.3 | 35.3 | 31.1 | 30.9 |

Table 4: Quantization results of larger-scale instruction-tuned models, including models of 123B and 405B. This table reports zero-shot accuracy (↑) on ARC and MMLU benchmarks.

| Model. | Prec. | Method | ARC-e | ARC-c | MMLU | | | | |
|---|---|---|---|---|---|---|---|---|---|
| | | | | | STEM | Hums | Social | Others | Avg. |
| Mistral-Large-123B-Instruct | W4A16 | LeanQuant | 85.1 | 64.0 | 76.6 | 77.3 | 89.2 | 85.9 | 82.3 |
| | | GPTQ | 84.6 | 64.0 | 76.3 | 77.2 | 89.3 | 85.2 | 82.0 |
| | | OSAQ$_{+GPTQ}$ | 85.0 | 64.1 | 76.7 | 77.4 | 89.3 | 85.7 | 82.3 |
| Llama-3.1-405B-Instruct | W4A16 g128 | LeanQuant | 88.3 | 64.8 | 82.7 | 83.2 | 90.6 | 87.7 | 86.1 |
| | | GPTQ | 88.2 | 65.1 | 82.3 | 82.6 | 90.5 | 87.5 | 85.7 |
| | | OSAQ$_{+GPTQ}$ | 88.3 | 65.0 | 82.6 | 83.2 | 90.8 | 87.7 | 86.1 |

scale, OSAQ remains effective. On Mistral-Large-123B-Instruct, OSAQ+GPTQ achieves an average MMLU score of 82.3%, matching the performance of LeanQuant. On LLaMA-3.1-405B-Instruct, OSAQ+GPTQ attains an average of 86.1% on MMLU, outperforming vanilla GPTQ.

**Evaluation with Quantized Weights and KV-Cache.** Table 5 reports perplexity results when both weights and KV-Cache are quantized to 4-bits. Beyond the weight-only scenario, this experiment validates OSAQ in another practical configuration that also quantizes the KV-Cache. For LLaMA2-7B, perplexity on WikiText2 decreases from 5.64 to 5.59 and on C4 from 7.49 to 7.43, while for LLaMA2-13B it decreases from 5.00 to 4.95 on WikiText2 and from 6.89 to 6.81 on C4.

Table 5: Quantization results under the 4-bit weights and 4-bit KV-Cache setting. This table reports perplexity results (↓) of LLaMA2 models of different scales.

| Prec. | Method | WikiText2 | | C4 | |
|---|---|---|---|---|---|
| | | 2-7B | 2-13B | 2-7B | 2-13B |
| FP16 | Baseline | 5.47 | 4.88 | 6.97 | 6.46 |
| W4A16KV4 | OmniQuant | 6.09 | 5.18 | 8.98 | 7.30 |
| | WKVQuant | 5.64 | 5.00 | 7.49 | 6.89 |
| | OSAQ$_{+WKVQuant}$ | 5.59 | 4.95 | 7.43 | 6.81 |

## 6.3 ABLATION STUDIES

**Impact of Additive Transformation on FP Model.** Table 6 shows that additive transformation has only a negligible impact on FP16 models (e.g., from 5.47 to 5.52 on WikiText2 for LLaMA2-7B), while bringing clear gains under quantization. For instance, GPTQ+Add. reduces perplexity from 8.37 to 6.75 on Wiki-Text2. This confirms that the additive transformation preserves FP performance while significantly improving low-bit quantization.

Table 6: Impact of additive transformation on original FP model performance.

| Prec. | Method | WikiText2 | | C4 | |
|---|---|---|---|---|---|
| | | 2-7B | 2-13B | 2-7B | 2-13B |
| FP16 | Baseline | 5.47 | 4.88 | 6.97 | 6.46 |
| | Baseline+Add. | 5.52 | 4.95 | 7.01 | 6.54 |
| W3A16 | GPTQ | 8.37 | 6.44 | 9.81 | 8.02 |
| | GPTQ+Add. | 6.75 | 5.72 | 8.70 | 7.54 |

**Effect of Softmax-$\infty$ Approximation.** Table 7 compares the effect of Softmax-$\infty$ approximation with directly applying $\ell_2$ norm. Direct $\ell_2$ norm optimization is less effective in suppressing outliers, leading to higher perplexity. In contrast, Softmax-$\infty$ approximation, which serves as a differentiable proxy of the $\ell_\infty$ norm, achieves substantially lower perplexity.

Table 7: Effect of Softmax-$\infty$ approximation compared to directly using $\ell_2$ norm.

| Prec. | Method | WikiText2 | | C4 | |
|---|---|---|---|---|---|
| | | 2-7B | 2-13B | 2-7B | 2-13B |
| FP16 | Baseline | 5.47 | 4.88 | 6.97 | 6.46 |
| W3A16 | $\ell_2$ norm | 7.82 | 6.11 | 9.13 | 7.88 |
| | Softmax-$\infty$+$\ell_2$ norm | 6.75 | 5.72 | 8.70 | 7.54 |

**Selection of Hyperparameters.** Figure 5 shows grid search results of four hyperparameters under 3-bit quantization for LLaMA2. $\gamma$ controls null space size, $\tau$ adjusts outlier emphasis in the Softmax-$\infty$ approximation, $\mu_1$ regulates additive strength, and $\mu_2$ suppresses bias shifts. The results remain stable across different values, demonstrating robustness to hyperparameter choice.

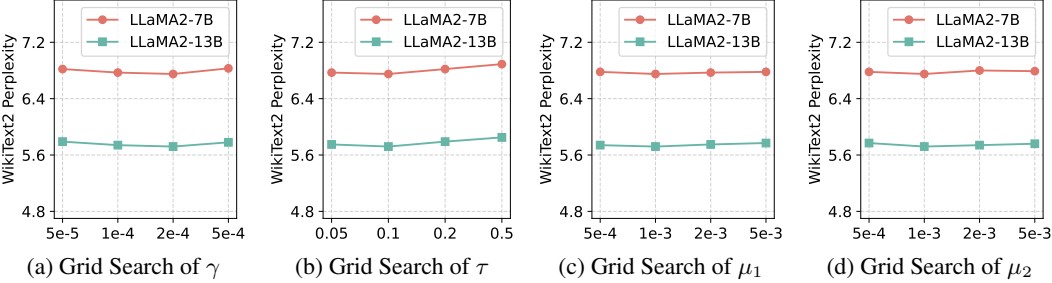

(a) Grid Search of $\gamma$     (b) Grid Search of $\tau$     (c) Grid Search of $\mu_1$     (d) Grid Search of $\mu_2$

Figure 5: Grid search of hyperparameters for LLaMA2 models under the 3-bit quantization setting. The results show that the quantization performance is robust to the choice of hyperparameters.

**Quantization Runtime.** Table 8 reports the quantization runtime on an Nvidia A100 GPU. Compared with GPTQ, OSAQ+GPTQ incurs only a marginal overhead (e.g., 24 min vs. 22 min on LLaMA2-7B). This is because the proposed OSAQ obtains the coefficient matrix in closed form, without requiring iterative optimization or additional training.

Table 8: Quantization runtime on an Nvidia A100 GPU. As shown, OSAQ incurs only a marginal overhead.

| Method | 2-7B | 2-13B | 2-70B |
|---|---|---|---|
| GPTQ | 22 min | 40 min | 4.0 hr |
| OSAQ+GPTQ | 24 min | 45 min | 4.6 hr |

## 7 CONCLUSION

In this work, we introduced OSAQ, an outlier self-absorption method based on additive transformation for low-bit weight-only quantization of LLMs. By exploiting the second-order low-rank property of the Hessian, OSAQ identifies a stable null space and constructs an additive transformation that effectively suppresses weight outliers while preserving task loss. Unlike multiplicative approaches, the transformation is fully absorbed into the weights, requiring no adjustment of neighboring layers and incurring no additional inference cost. Furthermore, the construction coefficients are obtained in closed form, making it free from training or iterative optimization. Extensive experiments across multiple models and tasks demonstrate the effectiveness of OSAQ.

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

# APPENDIX

## A MORE VISUALIZATION RESULTS OF WEIGHT DISTRIBUTIONS

Here, we provide additional visualization results to further illustrate the effect of the proposed method. Figure 6 presents the weight distributions in the attention module of the 0-th layer of LLaMA2-7B, inluding k_proj, v_proj, q_proj, and o_proj, while Figure 7 shows the corresponding results in the MLP module, including up_proj, gate_proj, and down_proj.

In both cases, the original model exhibits prominent outliers that significantly enlarge the weight range, which in turn increases the difficulty of representing weights under low-bit quantization. By approximating the $\ell_\infty$ norm in the coefficient optimization, OSAQ achieves strong suppression of these outliers, effectively tightening the distribution and reducing the numerical range. As a result, the overall weight distribution becomes smoother and more regular, making the quantization process more stable and yielding better quantization performance.

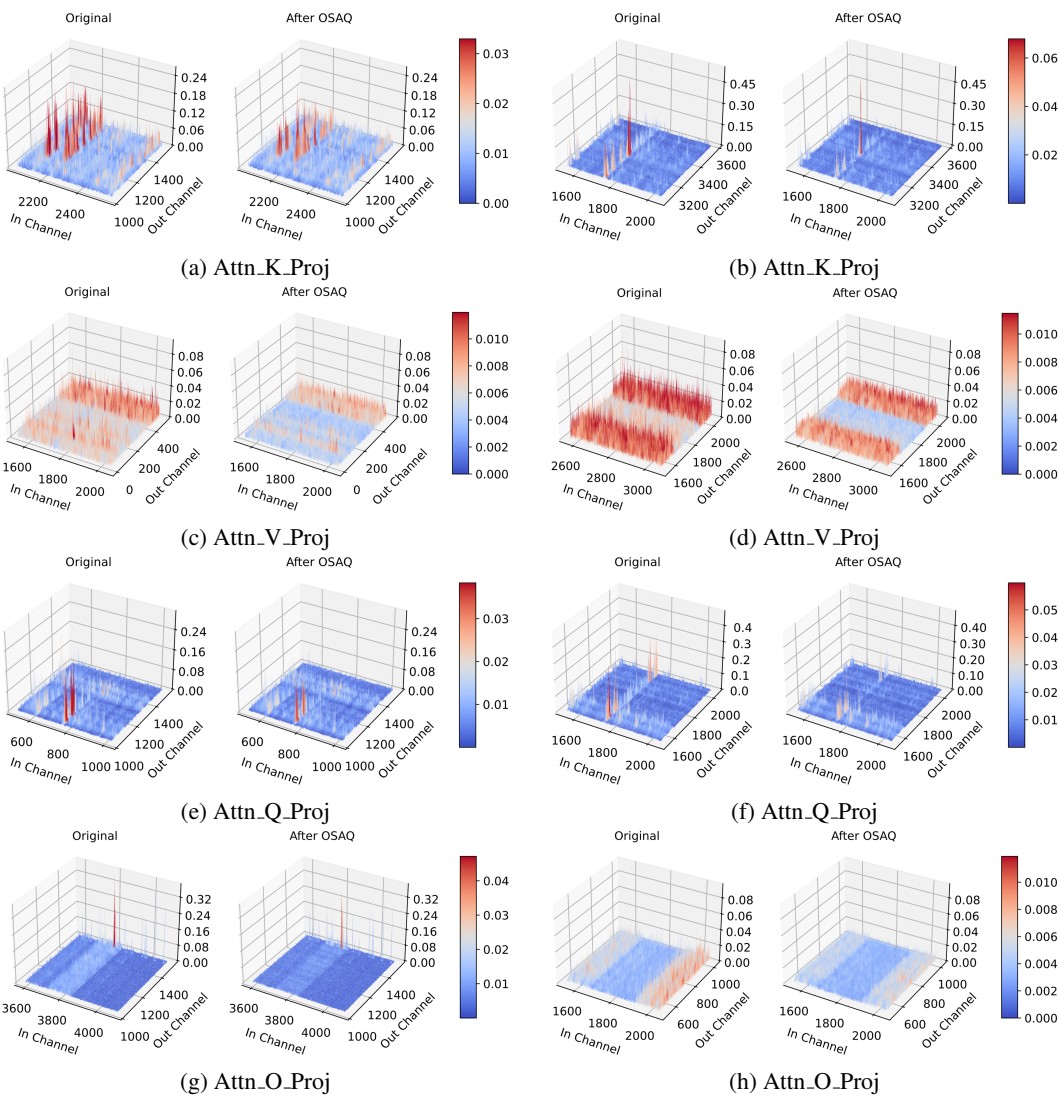

Figure 6: Weight distributions before and after the additive transformation in 0-th layer's attention module of LLaMA2-7B. The original model exhibits prominent outliers, whereas OSAQ effectively suppresses them, substantially reducing the difficulty of quantization.

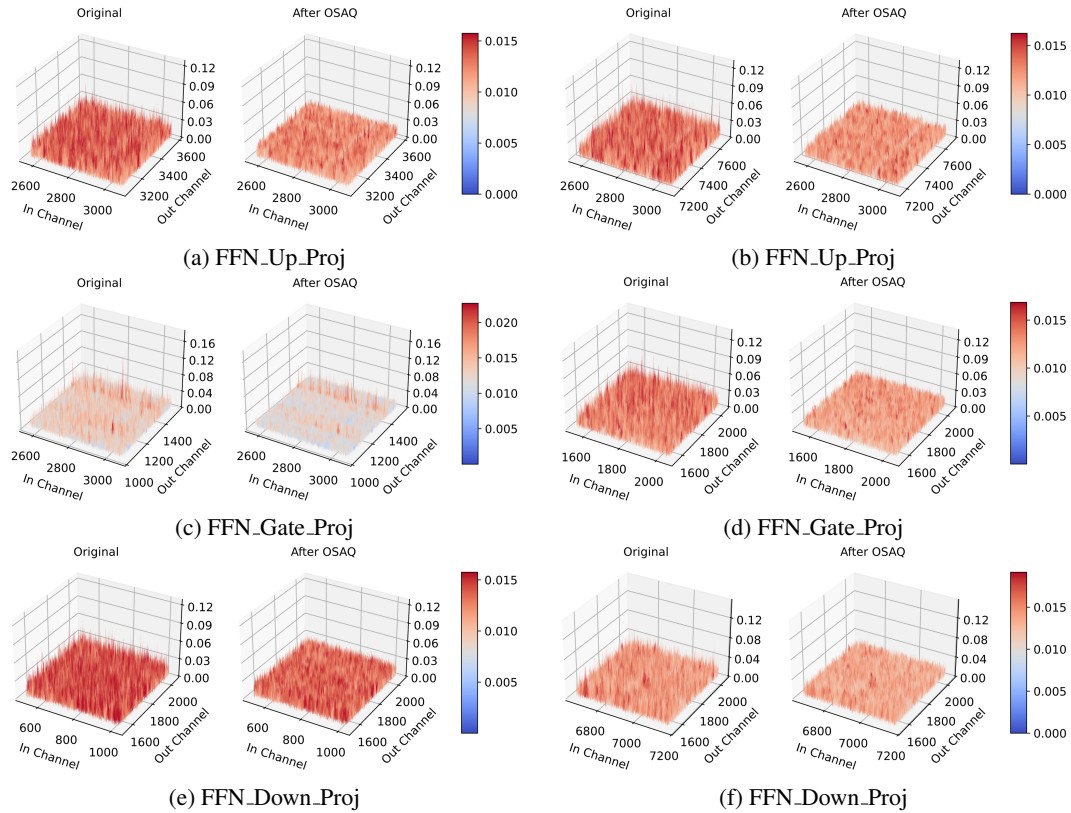

Figure 7: Weight distributions before and after the additive transformation in 0-th layer's MLP module of LLaMA2-7B. The original model exhibits prominent outliers, whereas OSAQ effectively suppresses them, substantially reducing the difficulty of quantization.

## B ASSESSMENT OF INFERENCE SPEEDUP

Table 9 reports the per-token generation latency of different models during the decoding stage on an Nvidia A100 GPU. As expected, quantization substantially reduces inference time compared to the FP16 baseline, and W4A16 quantization yields speedups of $1.89\times$, $2.41\times$, and $1.96\times$ on LLaMA2-7B, LLaMA2-13B, and LLaMA3-8B, respectively. Importantly, OSAQ introduces no additional inference overhead, ensuring that the acceleration ratios remain consistent with standard baselines.

Table 9: Per-token generation latency for the decoding stage on an Nvidia A100 GPU. As shown, OSAQ does not introduce any additional inference overhead, thus maintaining consistent acceleration ratios.

| Method | 2-7B | 2-13B | 3-8B |
|---|---|---|---|
| FP16 latency | 10.8 ms | 19.1 ms | 12.4 ms |
| W4A16 latency | 5.71 ms | 7.90 ms | 6.29 ms |
| W4A16 Speedup | $1.89\times$ | $2.41\times$ | $1.96\times$ |

