# OpenReview forum: "OSAQ: Outlier Self-Absorption for Accurate Low-bit LLM Quantization"
_ICLR.cc/2026/Conference — Submitted to ICLR 2026_

### Official Review · Reviewer_PzeR · 2025-10-25

**Soundness:** 2
**Presentation:** 2
**Contribution:** 2
**Rating:** 4
**Confidence:** 4

**Summary:**

The authors propose a method to mitigate outliers in the weights of LLMs. They first observe that the null space of the loss Hessian is relatively stable. Building on this, they add $\Delta W$ within that null space to minimize $l-\infty$ of $W+\Delta W$. They further approximate the objective utilizing the Softmax-$\infty$ Objective Appoximation and derive a closed-form solution. Experiments verify its effectiveness on the LLaMA-series models in terms of PPL, QA tasks.

**Strengths:**

1. The motivation is clear and strong, and the proposed method is established based on the observation.
2. The paper is well-organized overall, making it easy to follow.
3. The experiment and visualization verify the effectiveness of the proposed method.

**Weaknesses:**

1. The settings and detailed visualization procedure used in Figure 1(b) remain unclear. The authors should elaborate to clarify the observation. The norm length appears odd in the 2D representation.
2. It seems somewhat contradictory that the null space of  $H$ is obtained via the null space $X^TX$, yet the authors claim that the null space of $X$ is not consistent in Figure 1(b).
3. Sensitivity/robustness of approximation to obtain the null space with using approximation method or not ($X^TX$ vs $H$), the threshold, the size of the calibration set, and/or the chosen subspace dimension should be reported.
4. The gains appear limited. Across models and bit widths, the improvements are not consistently stable or significant.
5. The authors mentioned the assessment on MTBench in Line 296, but I cannot find any results regarding it.

**Questions:**

1. Can the proposed method be applied to weight–activation quantization settings? Rotations may change inter-layer activation inputs, which could make implementing $W+\Delta W$ harder.
2. What is the performance of solely use OSAQ (i.e. OSAQ+RTN) to quantize LLMs? To what extent does the method rely on other quantization methods?

---

> ### Author Response · Authors · 2025-11-19
>
> Thank you for your careful review and constructive insights. We provide detailed replies below and hope they clarify the points you raised.
>
> ---
>
> > W1: The settings and detailed visualization procedure used in Figure 1(b) remain unclear. The authors should elaborate to clarify the observation. The norm length appears odd in the 2D representation.
> >
>
> Thanks for your great comment.
> We provide a more detailed explanation of the setup in Figure 1(b). Each vector in the null space is high-dimensional (e.g., 4096 dimensions). For visualization, we project these vectors into a 2D space corresponding to the horizontal axis (Proj Dim 1) and vertical axis (Proj Dim 2). Thus, each point in the figure represents a vector, and **what is visualized is its direction** (Dim 1, Dim2) in the 2D projected space.
> To make the comparison clearer, we would like to clarify the following details:
> - Typically, each vector should be visualized as an arrow from the origin to the corresponding point.
> However, since there are many vectors, drawing all arrows would make the figure cluttered. Therefore, the plot **shows only the endpoints** (points), omitting the line segments from the origin.
> - In Figure 1(b), our focus is on comparing the consistency or variation of the **directions of null-space** vectors under different inputs.
> To make the directional relationships more readable, we **manually normalized the vectors** of different samples to different lengths, so the visualization highlights whether their directions overlap. For vectors that share a common direction in the null space, we draw dashed lines to indicate that they correspond to the same underlying direction.
>
>
> > W2: It seems somewhat contradictory that the null space of X is obtained via the null space X^T X, yet the authors claim that the null space of X is not consistent in Figure 1(b).
> >
>
> Thanks for your insightful comment.
> To avoid computing the Hessian H directly, we follow prior work such as GPTQ [1] and use X^T X as a substitute. Here, we explain the relationship between the null spaces of X^T X and X, and later compare X^T X with H in W3.
>
> First, as stated in line 229, neither X nor H has eigenvalues that are exactly zero. Therefore, we define the null space using cumulative tail energy, i.e., by collecting the eigenvectors associated with extremely small eigenvalues.
> Based on this, we explain why the null spaces of X^T X and X differ:
> - X^T X squares the singular values, which **amplifies the separation** between large and small components. A tiny singular value λ (e.g., 10^-7) becomes λ^2, making the ordering of small singular values more distinguishable and less likely to be swapped due to noise.
> - X^T X is a second-order statistic, meaning it **averages out sample-specific fluctuations**. Random or instantaneous variations present in individual samples cancel out in X^T X, leading to significantly more stable directions.
>
> [1] Frantar, Elias, et al. "OPTQ: Accurate post-training quantization for generative pre-trained transformers." International Conference on Learning Representations. 2023.

---

> > ### Author Response · Authors · 2025-11-19
> >
> > > W3: Sensitivity/robustness of approximation to obtain the null space with using approximation method or not (X^T X vs H), the threshold, the size of the calibration set, and/or the chosen subspace dimension should be reported.
> > >
> >
> > Thanks for your insightful comment.
> > Following your suggestion, we conducted comparisons between X^T X and H from several perspectives, including the threshold, the size of the calibration set, and the chosen subspace dimension, to further validate the effectiveness of using X^T X as a substitute for H. Here, H is computed using torch.func.hessian.
> >
> > (1) First, we examine the relationship between the magnitudes of the eigenvalues of X^T X and H, in order to understand differences in their thresholds. Similar to Figure 1, using data from the layer0.attn.k proj module in LLaMA2-7B, we report the cumulative energy ratio of the largest 3000 eigenvalues out of 4096:
> >
> > | Method   | Cumulative Energy Ratio |
> > |----------|--------------------------|
> > | X^T X    | 99.9914%                 |
> > | H        | 99.9944%                 |
> >
> >
> > These results indicate that the **threshold selection can be consistent** between the two.
> >
> > (2) Next, we evaluate the effect of different calibration set sizes, [64, 128, 256, 512, 1024], on X^T X and H. The W3A16 quantization results of GPTQ+OSAQ are shown below:
> >
> > | Model      | Method | Calibration size | WikiText2 | C4   |
> > |------------|--------|------------------|-----------|------|
> > | LLaMA2-7B  | H      | 64               | 6.81      | 8.80 |
> > |         |        | 128              | 6.73      | 8.69 |
> > |          |        | 256              | 6.72      | 8.67 |
> > |            |        | 512              | 6.70      | 8.67 |
> > |            |        | 1024             | 6.69      | 8.66 |
> > |            | X^T X  | 64               | 6.84      | 8.81 |
> > |            |        | 128              | 6.75      | 8.70 |
> > |            |        | 256              | 6.74      | 8.70 |
> > |            |        | 512              | 6.72      | 8.69 |
> > |            |        | 1024             | 6.72      | 8.69 |
> >
> > From these results, we observe:
> > - In our main experiments, we use a calibration size of 128. Reducing it to 64 leads to noticeable performance degradation. Increasing the size beyond 128 yields only minor improvements while significantly increasing computational cost, so **128 offers a good balance**.
> > - Directly using H provides slightly better final accuracy due to more precise computation. However, X^T X drastically reduces computation cost while achieving comparable performance, making it a **better trade-off** and more practical.
> >
> >
> > (3) Finally, we compare X^T X and H with respect to the chosen subspace dimension. Using data from the layer0.attn.k proj module in LLaMA2-7B, we compute the cosine overlap between the null spaces N1 and N2 by evaluating the singular values of N1^T N2. Values closer to 1 indicate nearly identical subspaces. The top-5 singular values are reported below:
> >
> > |    Rank       | 1     | 2     | 3     | 4     | 5     |
> > |---------------|-------|-------|-------|-------|-------|
> > | Singular values of N1^T N2 | 0.980 | 0.973 | 0.971 | 0.966 | 0.960 |
> >
> > This indicates that the directions of N1 and N2 are **highly consistent**.
> >
> > > W4: The gains appear limited. Across models and bit widths, the improvements are not consistently stable or significant.
> > >
> >
> > Thanks for your nice concern.
> > OSAQ introduces an additive transformation that departs from the existing multiplicative paradigm, providing a new perspective for handling weight outliers. As such, it serves as a plug-and-play module that can be combined with existing quantization methods to further mitigate weight outliers and improve performance. We would like to further clarify the effectiveness and practicality of OSAQ:
> > - **OSAQ is simple to implement and introduces minimal computational overhead during quantization**.
> > Thanks to the proposed closed-form solution, the entire pipeline requires neither iterations nor training, and includes no auxiliary or heuristic procedures. As a result, the computational cost added to the quantization process is very small.
> > - **OSAQ is compatible with various quantization techniques and introduces zero overhead during inference**.
> > Once quantized, the model's inference process remains unchanged, as OSAQ does not modify the runtime computation.
> > - **OSAQ consistently provides accuracy gains across models and datasets**.
> > For example, in W3A16 quantization of LLaMA-7B, OSAQ improves perplexity on WikiText2 by 1.62 compared with GPTQ.
> >
> > Therefore, OSAQ offers accuracy improvements at very low quantization cost, with no impact on inference efficiency. We believe this **effort-benefit ratio is valuable** for real-world applications.
> >
> > Moreover, since the additive transformation represents a new paradigm beyond scaling or rotation, this work is an initial exploration enabled by a new insight. We plan to further investigate and expand the potential of additive transformations in future work.

---

> ### Author Response · Authors · 2025-11-19
>
> > W5: The authors mentioned the assessment on MTBench in Line 296, but I cannot find any results regarding it.
> >
>
> Thanks for your nice comment.
>
> We apologize for the previously missing experimental data. The table below reports the MTBench results:
>
> | Prec.  | Method        | LLaMA2-7B | LLaMA2-13B |
> |--------|---------------|-----------|-------------|
> | FP16   | Baseline      | 3.83      | 4.69        |
> | W4A16  | GPTQ          | 3.66      | 4.50        |
> |        | OSAQ + GPTQ   | **3.72**      | **4.55**        |
> | W3A16  | GPTQ          | 3.37      | 4.32        |
> |        | OSAQ + GPTQ   | **3.47**      | **4.41**        |
>
>
>
> As shown, OSAQ consistently provides stable performance improvements.
>
> > Q1: Can the proposed method be applied to weight-activation quantization settings? Rotations may change inter-layer activation inputs, which could make implementing W+ΔW harder.
> >
>
> Thanks for your great question.
> We would like to clarify that OSAQ applies an additive perturbation to the weights to mitigate outliers, and, as stated in line 49, its primary goal is to address issues arising in **weight-only quantization**. Accordingly, in our experimental comparisons, we evaluate OSAQ against several widely used weight-only methods, including GPTQ, AWQ, and QuIP.
>
> In contrast, within weight-activation quantization, outliers in the activation distribution typically become the dominant performance bottleneck, which is why existing studies place greater **emphasis on activation handling**. Representative methods such as QuaRot, DuQuant, and SpinQuant are all designed specifically to address activation-side challenges.
>
> Despite this, following your suggestion, we additionally conducted experiments on weight-activation quantization, and the results on WikiText2 are summarized below:
>
>
> | Prec. | Method             | LLaMA2-7B | LLaMA2-13B |
> |-------|---------------------|-----------|------------|
> | FP16  | Baseline            | 5.47      | 4.88       |
> | W4A4  | QuaRot              | 6.10      | 5.40       |
> |       | OSAQ + QuaRot       | **6.03**  | **5.32**   |
> |       | DuQuant             | 6.28      | 5.42       |
> |       | OSAQ + DuQuant      | **6.19**  | **5.34**   |
> |       | SpinQuant           | 5.90      | 5.30       |
> |       | OSAQ + SpinQuant    | **5.84**  | **5.25**   |
>
> Regarding the experimental results above, we would like to provide the following additional clarifications:
> - In our implementation, OSAQ is used as a plug-and-play component and is always applied before other methods. That is, OSAQ first suppresses outliers to make the weights easier to quantize, and then the subsequent quantization method is applied. Therefore, any later operations, such as rotation, which may change inter-layer activation inputs, **do not affect the applicability or correctness of OSAQ**.
> - In weight-activation quantization, applying OSAQ to suppress weight outliers still **yields benefits**. On the one hand, the weight outliers themselves are mitigated; on the other hand, this preprocessing step can also implicitly facilitate the downstream weight-activation transformations performed by other methods.
> - Indead, since activation outliers are the dominant performance bottleneck in weight-activation quantization, **applying OSAQ only to weights** naturally results in limited performance improvement. In future work, we plan to investigate the possibility of extending the additive transformation to activations as well, achieving further gains in activation quantization.
>
>
> > Q2: What is the performance of solely use OSAQ (i.e. OSAQ+RTN) to quantize LLMs? To what extent does the method rely on other quantization methods?
> >
>
> Thanks for your nice question.
> Following your suggestion, we conducted OSAQ+RTN experiments on the LLaMA2-7B model, and the results are shown below:
>
> | Prec.        | Method       | WikiText2 | C4   |
> |--------------|--------------|-----------|------|
> | W4A16        | RTN          | 6.11      | 7.71 |
> |              | OSAQ + RTN   | **5.89**      | **7.54** |
> | W3A16 g128   | RTN          | 6.66      | 8.40 |
> |              | OSAQ + RTN   | **6.49**      | **8.25** |
> | W3A16        | RTN          | 539       | 402  |
> |              | OSAQ + RTN   | **8.82**      | **9.94** |
>
>
> As demonstrated, directly applying OSAQ on top of RTN also **yields consistent and significant improvements**. Moreover, we would like to emphasize that OSAQ does not involve any complex learning procedures nor introduce more advanced quantizers; therefore, it is not intended to be compared directly with end-to-end quantization methods. Instead, its role is to serve as a **plug-and-play component**, a complementary mechanism to scaling- or rotation-based approaches for mitigating outliers, providing additional performance gains.

---

### Official Review · Reviewer_sBRH · 2025-10-26

**Soundness:** 3
**Presentation:** 2
**Contribution:** 3
**Rating:** 6
**Confidence:** 4

**Summary:**

This paper addresses LLMs’ high inference resource use and latency via post-training weight-only quantization, tackling the key obstacle of weight outliers (poorly handled by existing scaling/rotation methods). It proposes OSAQ, which uses Hessian’s low-rank consistency across inputs to identify a stable null space. By linearly combining null-space vectors, OSAQ builds an additive weight transformation that suppresses outliers without task loss, absorbs offline (no inference overhead), and uses a closed-form solution (no heavy training). Experiments show OSAQ boosts low-bit quantization—e.g., 2-bit OSAQ+GPTQ cuts perplexity over 40% vs. vanilla GPTQ.

**Strengths:**

1. Starting from the perspective of weights, the authors perturb the weights using addition while maintaining an approximate invariance of the output, thereby smoothing the weight distribution. Their method is compatible with weight-calibrating approaches such as GPTQ, and the authors have verified the effectiveness of the method through extensive experiments.
2. The authors' method does not introduce any additional inference overhead, and only incurs 10% to 20% overhead during the calibration process.

**Weaknesses:**

I believe the authors' work is very rigorous and has no obvious weaknesses.

**Questions:**

1. Rotation smooths the distribution by transferring outliers from one channel to other channels, so I am confused why the authors claim that rotation cannot effectively eliminate outliers？
2. How does the authors' method perform in scenarios where activation values are also quantized?
3. Can the authors' method be combined with methods based on scale and rotation?

---

> ### Author Response · Authors · 2025-11-19
>
> Thank you very much for your positive feedback of our work and for the constructive comments. We provide detailed responses to your questions below.
>
> ---
>
> > Q1: Rotation smooths the distribution by transferring outliers from one channel to other channels, so I am confused why the authors claim that rotation cannot effectively eliminate outliers?
> >
>
> Thanks for your great comment.
> We apologize for the confusion. In the paper, including the Introduction and Related Work sections, we review existing paradigms for eliminating outliers, such as smoothing and rotation. We want to clarify that we fully acknowledge the effectiveness of smoothing and rotation in mitigating outliers. Our statement is that these methods may not **completely eliminate outliers**, which motivates us to propose an additive transformation as a **complementary mechanism to further improve** performance. As stated in line 140:
> "This approach complements existing inter-layer multiplicative transformations, providing a synergistic scheme that further enhances the performance of low-bit LLM quantization."
>
>
> > Q2: How does the authors' method perform in scenarios where activation values are also quantized?
> >
>
> Thanks for your thoughtful question.
> We would like to clarify that OSAQ applies an additive perturbation to the weights to mitigate outliers, and, as stated in line 49, its primary goal is to address issues arising in **weight-only quantization**. Accordingly, in our experimental comparisons, we evaluate OSAQ against several widely used weight-only methods, including GPTQ, AWQ, and QuIP.
>
> In contrast, within weight-activation quantization, outliers in the activation distribution typically become the dominant performance bottleneck, which is why existing studies place greater **emphasis on activation handling**. Representative methods such as QuaRot, DuQuant, and SpinQuant are all designed specifically to address activation-side challenges.
>
> Despite this, following your suggestion, we additionally conducted experiments on weight-activation quantization, and the results on WikiText2 are summarized below:
>
>
> | Prec. | Method             | LLaMA2-7B | LLaMA2-13B |
> |-------|---------------------|-----------|------------|
> | FP16  | Baseline            | 5.47      | 4.88       |
> | W4A4  | QuaRot              | 6.10      | 5.40       |
> |       | OSAQ + QuaRot       | **6.03**  | **5.32**   |
> |       | DuQuant             | 6.28      | 5.42       |
> |       | OSAQ + DuQuant      | **6.19**  | **5.34**   |
> |       | SpinQuant           | 5.90      | 5.30       |
> |       | OSAQ + SpinQuant    | **5.84**  | **5.25**   |
>
> Regarding the experimental results above, we would like to provide the following additional clarifications:
> - In our implementation, OSAQ is used as a plug-and-play component, and when combined with other methods, it is always applied before them. OSAQ first suppresses weight outliers, making the weights easier to quantize, and then the subsequent quantization method is applied. Therefore, OSAQ is **fully compatible** with all compared approaches.
> - In weight-activation quantization, applying OSAQ to suppress weight outliers still **yields benefits**. On the one hand, the weight outliers themselves are mitigated; on the other hand, this preprocessing step can also implicitly facilitate the downstream weight-activation transformations performed by other methods.
> - Indead, since activation outliers are the dominant performance bottleneck in weight-activation quantization, **applying OSAQ only to weights** naturally results in limited performance improvement. In future work, we plan to investigate the possibility of extending the additive transformation to activations as well, achieving further gains in activation quantization.
>
>
> > Q3: Can the authors' method be combined with methods based on scale and rotation?
> >
>
> Thanks for your nice comment.
> Following your suggestion, we conducted experiments combining OSAQ with scale-based and rotation-based methods. Specifically, we used AWQ for scaling and QuIP for rotation. The combination was applied in the following order:
> OSAQ → AWQ → QuIP.
> The experimental results are shown below:
>
> | Prec.       | Method                | LLaMA2-7B | LLaMA2-13B |
> |-------------|------------------------|-----------|-------------|
> | FP16        | Baseline               | 5.47      | 4.88        |
> | W3A16 g128  | AWQ + QuIP             | 6.05      | 5.20        |
> |             | OSAQ + AWQ + QuIP      | **6.00**      | **5.14**        |
> | W3A16       | AWQ + QuIP             | 6.30      | 5.21        |
> |             | OSAQ + AWQ + QuIP      | **6.23**      | **5.15**        |
>
>
> As shown, although AWQ+QuIP already suppress outliers effectively, there is still additional room for OSAQ to **further mitigate outliers** and thereby improve quantization performance.

---

### Official Review · Reviewer_2RrR · 2025-11-01

**Soundness:** 3
**Presentation:** 3
**Contribution:** 3
**Rating:** 6
**Confidence:** 3

**Summary:**

The paper proposes OSAQ, an additive outlier-suppression scheme guided by the Hessian null space, with a closed-form coefficient solver via a Softmax-∞ surrogate. The update is absorbed into weights and adds no inference cost, complementing scaling/rotation methods. Broad experiments on LLaMA-2/3 and instruction-tuned 123B/405B show notable low-bit gains, especially at 3-bit and 2-bit.

**Strengths:**

1. Novel additive paradigm grounded in low-rank consistent Hessian; clear second-order loss-invariance rationale.
2. Closed-form per-channel solution; plug-and-play with GPTQ/AWQ/QuIP; zero inference overhead.

**Weaknesses:**

1. The improvement on the existing methods is incremental.

2. How robust is the “stable Hessian null space” assumption across layers, models, and domains? Any quantitative measures (e.g., principal angles) across batches?

3. The article should include more thorough comparisons/combination with recent rotation families (e.g., DuQuant/QuaRot/SpinQuant) under matched settings.

**Questions:**

Please see the weakness.

1. How sensitive is performance to the temperature 𝜏? Any heuristic for layer-wise adaptive 𝜏?

2. What guided the Softmax-∞ surrogate choice for ℓ∞? Did you try p-norm annealing (e.g., p→∞) or direct max-margin formulations?

3. Ablations for stability of the null space vs. calibration set size (e.g., 64/256/1k samples).

---

> ### Author Response · Authors · 2025-11-19
>
> Thank you for your recognition of our work and your thorough review. Below, We answer all your questions with detailed explanations and clarifications.
>
> ---
>
> > W1: The improvement on the existing methods is incremental.
> >
>
> Thanks for your great comment.
> OSAQ introduces an additive transformation that departs from the existing multiplicative paradigm, providing a new perspective for handling weight outliers. As such, it serves as a plug-and-play module that can be combined with existing quantization methods to further mitigate weight outliers and improve performance. We would like to further clarify the effectiveness and practicality of OSAQ:
> - **OSAQ is simple to implement and introduces minimal computational overhead during quantization**.
> Thanks to the proposed closed-form solution, the entire pipeline requires neither iterations nor training, and includes no auxiliary or heuristic procedures. As a result, the computational cost added to the quantization process is very small.
> - **OSAQ is compatible with various quantization techniques and introduces zero overhead during inference**.
> Once quantized, the model's inference process remains unchanged, as OSAQ does not modify the runtime computation.
> - **OSAQ consistently provides accuracy gains across models and datasets**.
> For example, in W3A16 quantization of LLaMA-7B, OSAQ improves perplexity on WikiText2 by 1.62 compared with GPTQ.
>
> Therefore, OSAQ offers accuracy improvements at very low quantization cost, with no impact on inference efficiency. We believe this **effort-benefit ratio is valuable** for real-world applications.
>
> Moreover, since the additive transformation represents a new paradigm beyond scaling or rotation, this work is an initial exploration enabled by a new insight. We plan to further investigate and expand the potential of additive transformations in future work.
>
>
> > W2: How robust is the “stable Hessian null space” assumption across layers, models, and domains? Any quantitative measures (e.g., principal angles) across batches?
> >
>
> Thanks for your thoughtful comment.
> Following your suggestion, we conducted a quantitative analysis by examining the **principal angles** between the null spaces N1 and N2 obtained from two different input batches across multiple layers. The cosine of each principal angle corresponds to a singular value of
> N1^T N2; the closer the singular value is to 1, the more aligned the two subspaces are. The table below reports the maximum singular value for each layer:
>
> | Layer | Layer0                  |                  |                  | Layer7                  |                  |                  |
> |-------|--------------------------|------------------|------------------|--------------------------|------------------|------------------|
> | Module | attn.k_proj        | attn.q_proj | attn.v_proj | attn.k_proj        | attn.q_proj | attn.v_proj |
> | Max singular value of N1^T N2 | 0.973            | 0.981            | 0.975            | 0.966            | 0.970            | 0.965            |
>
>
> As shown, across different layers, the Hessian null spaces obtained from different inputs exhibit very **high consistency**, indicating that the null space remains stable.

---

> > ### Author Response · Authors · 2025-11-19
> >
> > > W3: The article should include more thorough comparisons/combination with recent rotation families (e.g., DuQuant/QuaRot/SpinQuant) under matched settings.
> > >
> >
> > Thanks for your nice comment.
> > We would like to clarify that OSAQ applies an additive perturbation to the weights to mitigate outliers, and, as stated in line 49, its primary goal is to address issues arising in **weight-only quantization**. Accordingly, in our experimental comparisons, we evaluate OSAQ against several widely used weight-only methods, including GPTQ, AWQ, and QuIP.
> >
> > In contrast, within weight-activation quantization, outliers in the activation distribution typically become the dominant performance bottleneck, which is why existing studies place greater **emphasis on activation handling**. Representative methods such as QuaRot, DuQuant, and SpinQuant are all designed specifically to address activation-side challenges.
> >
> > Despite this, following your suggestion, we additionally conducted experiments on weight-activation quantization, and the results on WikiText2 are summarized below:
> >
> >
> > | Prec. | Method             | LLaMA2-7B | LLaMA2-13B |
> > |-------|---------------------|-----------|------------|
> > | FP16  | Baseline            | 5.47      | 4.88       |
> > | W4A4  | QuaRot              | 6.10      | 5.40       |
> > |       | OSAQ + QuaRot       | **6.03**  | **5.32**   |
> > |       | DuQuant             | 6.28      | 5.42       |
> > |       | OSAQ + DuQuant      | **6.19**  | **5.34**   |
> > |       | SpinQuant           | 5.90      | 5.30       |
> > |       | OSAQ + SpinQuant    | **5.84**  | **5.25**   |
> >
> > Regarding the experimental results above, we would like to provide the following additional clarifications:
> > - In our implementation, OSAQ is used as a plug-and-play component, and when combined with other methods, it is always applied before them. OSAQ first suppresses weight outliers, making the weights easier to quantize, and then the subsequent quantization method is applied. Therefore, OSAQ is **fully compatible** with all compared approaches.
> > - In weight-activation quantization, applying OSAQ to suppress weight outliers still **yields benefits**. On the one hand, the weight outliers themselves are mitigated; on the other hand, this preprocessing step can also implicitly facilitate the downstream weight-activation transformations performed by other methods.
> > - Indead, since activation outliers are the dominant performance bottleneck in weight-activation quantization, **applying OSAQ only to weights** naturally results in limited performance improvement. In future work, we plan to investigate the possibility of extending the additive transformation to activations as well, achieving further gains in activation quantization.
> >
> >
> > > Q1: How sensitive is performance to the temperature 𝜏? Any heuristic for layer-wise adaptive 𝜏?
> > >
> >
> > Thanks for your nice question.
> > The parameter 𝜏 adjusts the degree of outlier emphasis in the Softmax-∞ approximation. Its grid search results on the LLaMA2-7B and LLaMA2-13B models are shown in Fig. 5(b). The results remain stable across different values, demonstrating the robustness of our method to this hyperparameter.
> >
> > Motivated by your question, we further investigated the layer-wise adaptivity of 𝜏. As indicated in Equation (9), a smaller 𝜏 places more emphasis on outliers. Therefore, we conducted a preliminary experiment where, within the candidate set [0.05,0.1,0.2,0.5], layers with more pronounced outliers, measured by larger max(∣W∣), were assigned smaller values of
> > 𝜏, whereas flatter weight distributions were assigned larger values. The W3A16 quantization results are as follows:
> >
> > | Model        | Method             | WikiText2 | C4    |
> > |--------------|--------------------|-----------|-------|
> > | LLaMA2-7B | GPTQ               | 8.37      | 9.81  |
> > |              | GPTQ + OSAQ        | 6.75      | 8.70  |
> > |              | GPTQ + OSAQ + Ada τ | **6.59**      | **8.61**  |
> > | LLaMA2-13B | GPTQ              | 6.44      | 8.02  |
> > |              | GPTQ + OSAQ        | 5.72      | 7.54  |
> > |              | GPTQ + OSAQ + Ada τ | **5.60**      | **7.38**  |
> >
> > These results **validate the feasibility of adaptive** 𝜏. Note that in this experiment, the choice was made via manually defined thresholds on max(∣W∣). We believe that designing a more principled automatic selection strategy is a promising direction and plan to explore it in future work.

---

> > > ### Author Response · Authors · 2025-11-19
> > >
> > > > Q2: What guided the Softmax-∞ surrogate choice for ℓ∞? Did you try p-norm annealing (e.g., p→∞) or direct max-margin formulations?
> > > >
> > >
> > > Thanks for your great question.
> > > The purpose of the Softmax-∞ approximation is to transform the ℓ∞-norm in the optimization objective of Equation (8) into an ℓ2-norm. This is because an ℓ2-norm formulation leads to a strictly **convex quadratic optimization problem**, whose optimum can be obtained in **closed form** by simply setting the derivative to zero. This yields the explicit solution of OSAQ, as shown in Equation (11).
> > >
> > > In contrast, for p-norm annealing or direct max-margin formulations, to the best of our knowledge (and we welcome corrections if this understanding is inaccurate), one generally has to rely on LP/QP-based numerical solvers or iterative optimization procedures, which **do not provide a closed-form solution**.
> > >
> > > > Q3: Ablations for stability of the null space vs. calibration set size (e.g., 64/256/1k samples).
> > > >
> > >
> > > Thanks for your insightful question.
> > > Following your suggestion, we evaluated the effect of different calibration set sizes [64,128,256,512,1024]. The W3A16 quantization results for GPTQ+OSAQ are shown below:
> > >
> > > | Model | Calibration size | WikiText2 | C4   |
> > > |-------|------------------|-----------|------|
> > > |  LLaMA2-7B  | 64               | 6.84      | 8.81 |
> > > |       | 128              | 6.75      | 8.70 |
> > > |       | 256              | 6.74      | 8.70 |
> > > |       | 512              | 6.72      | 8.69 |
> > > |       | 1024             | 6.72      | 8.69 |
> > >
> > >
> > > In our main experiments, we use a calibration set size of 128. As the results indicate, reducing the size to 64 leads to a noticeable performance drop, while increasing the calibration size beyond 128 yields only marginal improvements but introduces significantly higher computational cost. Therefore, **128 provides a better balance** between performance and efficiency.

---

### Official Review · Reviewer_A4Q1 · 2025-11-01

**Soundness:** 2
**Presentation:** 3
**Contribution:** 2
**Rating:** 4
**Confidence:** 3

**Summary:**

This paper proposes OSAQ, a Hessian-based additive transformation method for mitigating systematic outliers in low-bit weight quantization of LLMs. The authors claim that by leveraging the low-rank consistency of the Hessian and operating within its null space, one can add a closed-form, inference-free correction term that absorbs outlier effects without retraining or extra computational cost.

**Strengths:**

- Extending Hessian-based quantization ideas toward a closed-form additive formulation is conceptually interesting and novel.
- If the proposed null-space additive transformation indeed preserves task loss and can be absorbed into model weights, it would be a practically appealing solution.

**Weaknesses:**

-  The experiments do not include key state-of-the-art methods addressing outliers, such as QuaRot, DuQuant, and SpinQuant, under a unified evaluation setup.
-  The numerical stability of the closed-form solution, especially for large layers or block-wise quantization, is not discussed.

**Questions:**

- Under what conditions does the transformation guarantee that the first-order term is zero, ensuring loss invariance? Does this assume the model is at or near a local optimum?
- If the calibration and inference distributions differ, does the null-space property still hold? Could you bound or quantify the resulting error?
- What is its relationship with the trace-based selection strategy used in HAWQ-V2, and what are the corresponding advantages or disadvantages?

---

> ### Author Response · Authors · 2025-11-19
>
> We sincerely appreciate your review and the insightful comments on our work. We provide detailed responses below and hope they address your concerns.
>
> ---
>
> > W1: The experiments do not include key state-of-the-art methods addressing outliers, such as QuaRot, DuQuant, and SpinQuant, under a unified evaluation setup.
> >
>
> Thanks for your great comment.
> We would like to clarify that OSAQ applies an additive perturbation to the weights to mitigate outliers, and, as stated in line 49, its primary goal is to address issues arising in **weight-only quantization**. Accordingly, in our experimental comparisons, we evaluate OSAQ against several widely used weight-only methods, including GPTQ, AWQ, and QuIP.
>
> In contrast, within weight-activation quantization, outliers in the activation distribution typically become the dominant performance bottleneck, which is why existing studies place greater **emphasis on activation handling**. Representative methods such as QuaRot, DuQuant, and SpinQuant are all designed specifically to address activation-side challenges.
>
> Despite this, following your suggestion, we additionally conducted experiments on weight-activation quantization, and the results on WikiText2 are summarized below:
>
>
> | Prec. | Method             | LLaMA2-7B | LLaMA2-13B |
> |-------|---------------------|-----------|------------|
> | FP16  | Baseline            | 5.47      | 4.88       |
> | W4A4  | QuaRot              | 6.10      | 5.40       |
> |       | OSAQ + QuaRot       | **6.03**  | **5.32**   |
> |       | DuQuant             | 6.28      | 5.42       |
> |       | OSAQ + DuQuant      | **6.19**  | **5.34**   |
> |       | SpinQuant           | 5.90      | 5.30       |
> |       | OSAQ + SpinQuant    | **5.84**  | **5.25**   |
>
> Regarding the experimental results above, we would like to provide the following additional clarifications:
> - In our implementation, OSAQ is used as a plug-and-play component, and when combined with other methods, it is always applied before them. OSAQ first suppresses weight outliers, making the weights easier to quantize, and then the subsequent quantization method is applied. Therefore, OSAQ is **fully compatible** with all compared approaches.
> - In weight-activation quantization, applying OSAQ to suppress weight outliers still **yields benefits**. On the one hand, the weight outliers themselves are mitigated; on the other hand, this preprocessing step can also implicitly facilitate the downstream weight-activation transformations performed by other methods.
> - Indead, since activation outliers are the dominant performance bottleneck in weight-activation quantization, **applying OSAQ only to weights** naturally results in limited performance improvement. In future work, we plan to investigate the possibility of extending the additive transformation to activations as well, achieving further gains in activation quantization.
>
>
>
> > W2: The numerical stability of the closed-form solution, especially for large layers or block-wise quantization, is not discussed.
> >
>
> Thanks for your nice comment.
> We would like to clarify that in Equation (10), the optimization objective is formulated and solved **independently for each output channel**, meaning that the computation of 𝑏 is performed at the channel level. Then, in Equation (12), these per-channel results are aggregated to form 𝛽, rather than solving for the entire 𝛽 vector directly. This design aligns with the fact that quantization scales are computed on a per-output-channel basis, and therefore provides **strong scalability**, making the approach well suited for large layers or block-wise quantization.
>
> In addition, Table 4 reports quantization results on large-scale models (123B and 405B), both of which contain large layers, further demonstrating the effectiveness of our method in handling large-layer settings.

---

> > ### Author Response · Authors · 2025-11-19
> >
> > > Q1: Under what conditions does the transformation guarantee that the first-order term is zero, ensuring loss invariance? Does this assume the model is at or near a local optimum?
> > >
> >
> > Thanks for your thoughtful question.
> > As you correctly pointed out, for a well-trained pretrained model, standard gradient-based optimization methods (including SGD, Adam, and their variants) drive the parameters toward a local optimum of the training loss surface. Near such a point, the first-order gradient should theoretically **converge to zero or remain very close to zero**. Consequently, in the Taylor expansion of the error, the first-order term can be reasonably ignored.
> > This local-optimality assumption has been widely adopted in many classical quantization papers, such as [1][2].
> > [1] Nagel, Markus, et al. "Up or down? adaptive rounding for post-training quantization." International conference on machine learning. 2020.
> > [2] Li, Yuhang, et al. "BRECQ: Pushing the limit of post-training quantization by block reconstruction." International Conference on Learning Representations. 2021.
> >
> >
> >
> > > Q2: If the calibration and inference distributions differ, does the null-space property still hold? Could you bound or quantify the resulting error?
> > >
> >
> > Thanks for your nice question.
> > Following your suggestion, we sampled two batches of input data from **two different datasets**, WikiText2 and C4, and evaluated the discrepancy between the corresponding null spaces. To quantify this, we examined the **principal angles** between the null spaces N1 and N2 induced by the two input batches across different layers. The cosine of each principal angle corresponds to a singular value of N1^T N2; the closer these singular values are to 1, the more aligned the subspaces are. Below we report the largest singular value for each layer:
> >
> >
> > | Layer | Layer0                |                 |                 | Layer7                |                 |                 |
> > |-------|------------------------|-----------------|-----------------|------------------------|-----------------|-----------------|
> > | Module | attn.k_proj      | attn.q_proj | attn.v_proj | attn.k_proj      | attn.q_proj | attn.v_proj |
> > | Max singular value of N1^T N2 | 0.967           | 0.977           | 0.973           | 0.970           | 0.969           | 0.968           |
> >
> >
> >
> > As shown, across different layers, the Hessian null spaces exhibit very **high consistency** under different inputs. This indicates that even when the input samples come from different data distributions (i.e., different datasets), the null space remains stable.
> >
> >
> > > Q3: What is its relationship with the trace-based selection strategy used in HAWQ-V2, and what are the corresponding advantages or disadvantages?
> > >
> >
> > Thanks for your great comment.
> > Both HAWQ-V2 and OSAQ make use of Hessian information to assess how layer-wise weights influence the task loss. However, we would like to clarify that the purpose and usage of the Hessian in the two methods are fundamentally different.
> >
> > In HAWQ-V2, the Hessian trace is used to estimate the sensitivity of each layer's weights to perturbations, thereby determining the relative importance of different layers and assigning different bit-widths accordingly. The trace is computed as the sum of all eigenvalues, capturing a global measure of curvature.
> >
> > In contrast, OSAQ leverages the Hessian to obtain the null space of the additive perturbation for each layer's weights. The null space is derived directly from the zero eigenvalues of the original Hessian, without relying on the trace as an intermediate metric. This null-space information defines the subspace in which additive transformations preserve the loss locally.
> >
> > Thus, the two methods differ both in objective and mechanism:
> > - The Hessian trace reflects **global eigenvalue characteristics** for layer importance ranking (HAWQ-V2).
> > - OSAQ instead requires **local zero-eigenvalue structure**, focusing on the null space for loss-invariant additive transformations.

---

### Author Response · Authors · 2025-12-02
**Rebuttal Summary of Submission 4939**

We sincerely appreciate the reviewers' careful and professional comments. Below is a summary of our response:

First, we are pleased to see that **all reviewers** (A4Q1, 2RrR, sBRH, PzeR) **recognize the motivation and novelty** of our work in using additive transformations to suppress outliers. They also praised the practicality of our **closed-form solution** (A4Q1, 2RrR), the **plug-and-play nature** of our method (2RrR, sBRH), and its **zero inference overhead** (A4Q1, 2RrR, sBRH). In particular, reviewer sBRH commented that the "*work is very rigorous and has no obvious weaknesses*".

Second, reviewers also provided several insightful suggestions. Their concerns generally fall into three major themes, to which we have provided detailed responses through additional experiments and explanations. A summary follows (W: Weekness, Q: Question):

- *Experiments on weight-activation quantization* (A4Q1 W1, 2RrR W3, sBRH Q2, PzeR Q1):
Our method is proposed primarily for weight-only quantization and suppresses weight outliers via additive transformations. Nevertheless, since our method is applied before other quantization methods (PzeR Q1), it is also compatible with weight-activation quantization. In the rebuttal, we added experiments with QuaRot, DuQuant, and SpinQuant, demonstrating **consistent performance gains**. These improvements arise from both suppressing weight outliers and, potentially, facilitating activation transformations.
- *Stability of the Null Space* (A4Q1 Q2, 2RrR W2, 2RrR Q3, PzeR W2 & W3):
Through extensive experiments, especially quantitative evaluations using the maximum singular value, we have verified the stability of the Null Space under various conditions, such as differing data distributions (A4Q1 Q2), across layers, models, and domains (2RrR W2), varying calibration set sizes (2RrR Q3), and the comparison of using X^T X versus the Hessian (PzeR W2 & W3).
These results support that, in our method, the **Null Space is robust and stable** across different models and datasets.
- *Performance gains not very large* (2RrR W1, PzeR W4):
Since this work represents an initial exploration of a new additive paradigm beyond scaling or rotation, the performance gains are indeed not extremely large, though important improvements are still observed (e.g., in W3A16 quantization of LLaMA-7B, OSAQ improves perplexity on WikiText2 by 1.62, from 8.37 to 6.75, compared with GPTQ).
However, we believe this **effort-benefit ratio**, providing accuracy gains at very low quantization cost and with no impact on inference efficiency, is valuable for real-world deployment, and we will further investigate the potential of additive transformations in future work.

Although we did not have enough time to discuss fully with the reviewers, we believe the experiments and explanations in our rebuttal can  address their concerns. Finally, we again thank all reviewers for their comments and especially thank the ACs for their time and effort.

---

### Meta-Review · Area_Chair_TveX · 2025-12-22

**Summary:**

This paper proposes OSAQ (Outlier Self-Absorption Quantization) to address systematic outliers in low-bit post-training quantization (PTQ) for LLMs. The key insight is that, under a local second-order approximation of weight quantization, certain directions (approximately related to the Hessian / an input-correlation matrix) exhibit near “zero-curvature / stable subspace” behavior. Based on this, the authors construct an additive weight perturbation along that subspace to absorb outliers into the weights, thereby shrinking the value range while approximately preserving the loss. OSAQ is plug-and-play and requires no additional training/iterations. The paper reports gains when integrated into frameworks such as GPTQ/AWQ/QuIP across multiple model scales and tasks, and positions OSAQ as a complementary module to existing scaling/rotation-based methods.

The reviews show some disagreement. Positive reviewers appreciate the new angle of moving from “multiplicative transformations (scaling/rotation)” to an “additive transformation,” the closed-form solution and engineering practicality, and the broad experimental coverage. Negative reviewers mainly question: (i) whether comparisons with recent outlier/rotation-related works are sufficiently comprehensive, (ii) the verifiability of key assumptions and implementation details, and (iii) the sensitivity to hyperparameters / calibration sets / subspace selection and the stability of the gains.

The rebuttal adds several key experiments and clarifies the rationale behind the null-space approximation, which mitigates the major concerns to some extent.

**Reviewer Concerns:**

**A. Points that are partially addressed/mitigated in the rebuttal**

1. Key comparisons and combination experiments are expanded: This addresses concerns about missing comparisons with recent outlier/rotation-related methods or weight-activation quantization systems. The authors clarify that OSAQ primarily targets *weight-only* quantization, and add combination experiments with Quarot/DuQuant/SpinQuant under weight-activation settings, showing that OSAQ as a preprocessing module can still yield improvements in several settings.

2. Stronger clarification on null-space stability and implementation details: In response to questions such as whether the null space varies with inputs, how Figure 1 is obtained, and whether the approximation is self-consistent, the rebuttal adds principal-angle consistency analyses across different batches/data distributions, and explains the motivation for the Hessian approximation as well as the strategies for choosing thresholds/subspace dimensionality.

3. Sensitivity analyses and missing metrics: The authors add sensitivity studies on calibration-set size and thresholds/hyperparameters, and include the previously missing MT-Bench results, reducing the risk that conclusions rely on incomplete experiments.



**B. Remaining concerns**

1. The magnitude and stability of gains remain borderline
   While the additive-transformation perspective is novel, some reviewers note that the improvements over existing quantization frameworks are still incremental and not consistently stable in certain settings. The rebuttal broadens the coverage, but evidence that gains are consistently significant across all mainstream setups/layer types/bit-widths is still not fully sufficient.

1. The theoretical narrative and assumption boundaries should be stated more cautiously
   The paper’s theoretical support hinges on null-space stability and an approximate loss-invariance derivation, which relies on a second-order approximation and an approximate Hessian. The rebuttal provides some empirical support, but the final version would benefit from more clearly specifying: the applicability conditions of the theoretical conclusions, the impact of approximation error, and whether changes in layer structure or calibration distribution could break the assumption.

**Reviewer Scores:**

The four reviewers’ scores show disagreement, and the two lower-score reviewers raise concrete and verifiable concerns. The rebuttal adds several experiments and explanations, substantially reducing the main risks related to missing critical evidence and clarifications.

Meanwhile, the positive reasons from the two higher-score reviewers, i.e., the methodological novelty, closed-form solution and engineering practicality, also become easier to defend after the rebuttal.

Overall, I expect the lower-score reviewers may slightly increase their scores into the borderline range. However, core concerns still warrant further clarification.

---

### Decision · Program_Chairs · 2026-01-26

Reject